# Definitive engineering strength and fracture toughness of graphene through on-chip nanomechanics

Sahar Jaddi [1] ✉, M. Wasil Malik[2,7], Bin Wang[2,3], Nicola M. Pugno [4,5], Yun Zeng[3], Michael Coulombier [1], Jean-Pierre Raskin[2] & Thomas Pardoen [1,6]

Fail-safe design of devices requires robust integrity assessment procedures which are still absent for 2D materials, hence affecting transfer to applications. Here, a combined on-chip tension and cracking method, and associated data reduction scheme have been developed to determine the fracture toughness and strength of monolayer-monodomain-freestanding graphene. Myriads of specimens are generated providing statistical data. The crack arrest tests provide a definitive fracture toughness of 4.4 MPa$\sqrt{\mathrm{m}}$. Tension on-chip provides Young's modulus of 950 GPa, fracture strain of 11%, and tensile strength up to 110 GPa, reaching a record of stored elastic energy ~6 GJ m$^{-3}$ as confirmed by thermodynamics and quantized fracture mechanics. A ~1.4 nm crack size is often found responsible for graphene failure, connected to 5-7 pair defects. Micron-sized graphene membranes and smaller can be produced defect-free, and design rules can be based on 110 GPa strength. For larger areas, a fail-safe design should be based on a maximum 57 GPa strength.

2D materials such as graphene have received attention in the context of a variety of potential applications driven by their exceptional properties[1]. Generally, 2D materials have a combined functional–structural role. However, the establishment of robust procedures that guarantee fail-safe designs for long-term reliable operation is essentially missing in the world of 2D materials. There is no doubt that the extreme strength (>100 GPa) and excellent ductility (>20%) predicted theoretically[2–4] and verified experimentally[5,6] on nanoscale specimens is a major asset. Nevertheless, literature studies are essentially limited to extremely small specimens, with limited statistical significance in terms of defect distribution and the number of test samples. In real applications involving wide 2D materials with a single or few layers, the failure resistance is dictated more by the population of defects than by the intrinsic strength and ductility. The literature mainly focuses on the fundamental mechanisms controlling the strength variations when contrasting defective and perfect lattice

structures[7–12]. The context is exactly similar to brittle-type materials with covalent or ionic bonds, such as for silicon with a theoretical strength of >25 GPa and fracture strain >20%[13] that, at the macroscopic wafer level, drops down to ~300 MPa and ~0.2%, respectively[14]. Reconciliation between theoretical and in-use values comes through fracture mechanics theory assuming a population of crack-type defects and Weibull analysis[15]. However, this approach requires generating reliable fracture toughness data with statistical value and the determination of the nature of the defects. This leads to major challenges in the context of 2D materials such as graphene.

Several approaches have been proposed to determine the mechanical properties of 2D materials. Deflection of suspended graphene membranes has generally been performed using an atomic force microscope tip[6], which, when applied to chemical vapor deposition (CVD)-graphene indicated a strength reduction by 40% due to grain boundaries[5]. The strength decreases with decreasing grain

[1]Institute of Mechanics, Materials and Civil Engineering, UCLouvain, Belgium. [2]Institute of Information and Communication Technologies, Electronics and Applied Mathematics, UCLouvain, Belgium. [3]School of Physics and Electronics, Hunan University, Changsha, China. [4]Laboratory for Bioinspired, Bionic, Nano, Meta Materials & Mechanics, Department of Civil, Environmental and Mechanical Engineering, University of Trento, Trento, Italy. [5]School of Engineering and Material Science, Queen Mary University of London, London, UK. [6]WEL Research Institute, Avenue Pasteur, 6, 1300 Wavre, Belgium. [7]Deceased: M. Wasil Malik. ✉e-mail: jaddi.sahar@gmail.com

boundary disorientation up to ~59%[16]. Burst testing[17] leads to a wide distribution of failure pressure, attributed to defects, slacks, and/or wrinkles in graphene. Nonetheless, applying uniaxial tension conditions on large-area crack-free graphene remains the most direct way for extracting representative strength data. Recently, in-situ push-to-pull tensile testing[17] performed on CVD-grown single-layer graphene (SLG) delivered a tensile strength of ~60 GPa with a maximum fracture strain of ~6% for ~10 μm² membrane. None of these methods provides fracture toughness values.

Determination of the mode I fracture toughness $K_{Ic}$ requires going a step further with the introduction of a sharp pre-crack and a method to estimate exactly when cracking initiates. The first calculation of the fracture toughness of graphene dates back to 2004 by Pugno and Ruoff[18] reporting a value $K_{Ic}$ = 3.2 or 3.45 MPa$\sqrt{m}$ for zigzag or armchair crack, respectively. Zhang et al. [19] pioneered in situ scanning electron microscope (SEM) fracture experiments on notched CVD-graphene. The specimens were mostly bilayer graphene (BLG) with a mean $K_{Ic}$ = 4 ± 0.6 MPa$\sqrt{m}$. In another study, $K_{Ic}$ of a 10-layer graphene is equal to 12 ± 3.9 MPa$\sqrt{m}$, much higher than for SLG[20]. Cao et al.[21] followed a similar approach using push-to-pull tensile testing to study the fracture behavior of pristine BLG[22], with $K_{Ic}$ ≈ 29.5 MPa$\sqrt{m}$. This is considerably larger than the previously reported values, and attributed to the nonlinearity of graphene[23]. The fracture of SLG was also studied by bulge testing[24], giving $K_{Ic}$ = 10.7 ± 3.3 MPa$\sqrt{m}$, disclosing also the environmental cracking susceptibility of CVD-graphene. Computational studies predict $K_{Ic}$ = 3–4 MPa$\sqrt{m}$ by molecular dynamics simulations[9] and $K_{Ic}$ = 3.1 – 4.5 MPa$\sqrt{m}$ by molecular mechanics simulations[25]. The cracking process has been qualitatively investigated by transmission electron microscopy (TEM)[26,27] revealing tearing directions mostly dictated by either the zigzag or armchair directions and with limited anisotropy. All the aforementioned experimental methods are affected by the problem of introducing a sharp pre-crack, by focused ion beam damage, and by the difficulty of generating statistical data, hence not delivering foundations for a true fail-safe design.

The objective of this study is to unravel the fracture resistance of SLG by combining an original crack-on-chip (COC) and a uniaxial tension-on-chip (TOC), namely (TOCOC) experimental approach to deliver definitive statistically representative data, from which a mechanical analysis can pave the way toward fail-safe design of 2D materials-based components/devices. In an effort to overcome the shortcomings encountered in the available approaches, several challenges have been addressed in this study: (i) the specimens were tested on-chip to circumvent the gripping, clamping, and transfer problems; (ii) the specimen shape was accurately controlled through lithography methods controlled and the geometrical dimensions can be measured with precision; (iii) an on-chip loading relying on a residual stress actuation principle was adapted to avoid the use of any external macroscopic or microscopic device as well as the associated alignment issues; (iv) many samples were produced and tested simultaneously for statistical analysis; (v) different sample sizes and shapes were produced to verify if the size dependence of the fracture resistance related to the defects population can be rationalized; (vi) the crack arrest principle adopted to circumvent the artifact of a blunted starter notch instead of a true pre-crack in the fracture mechanics sense required an adapted design.

## On-chip mechanical testing

The working principle of the on-chip testing method is to deform a graphene specimen by attaching it to a beam, which, when released from the substrate, contracts as a result of relaxation of internal tensile stress, thus imposing a displacement to the specimen. A schematic representation of both TOC and COC samples is shown in Fig. 1. In the

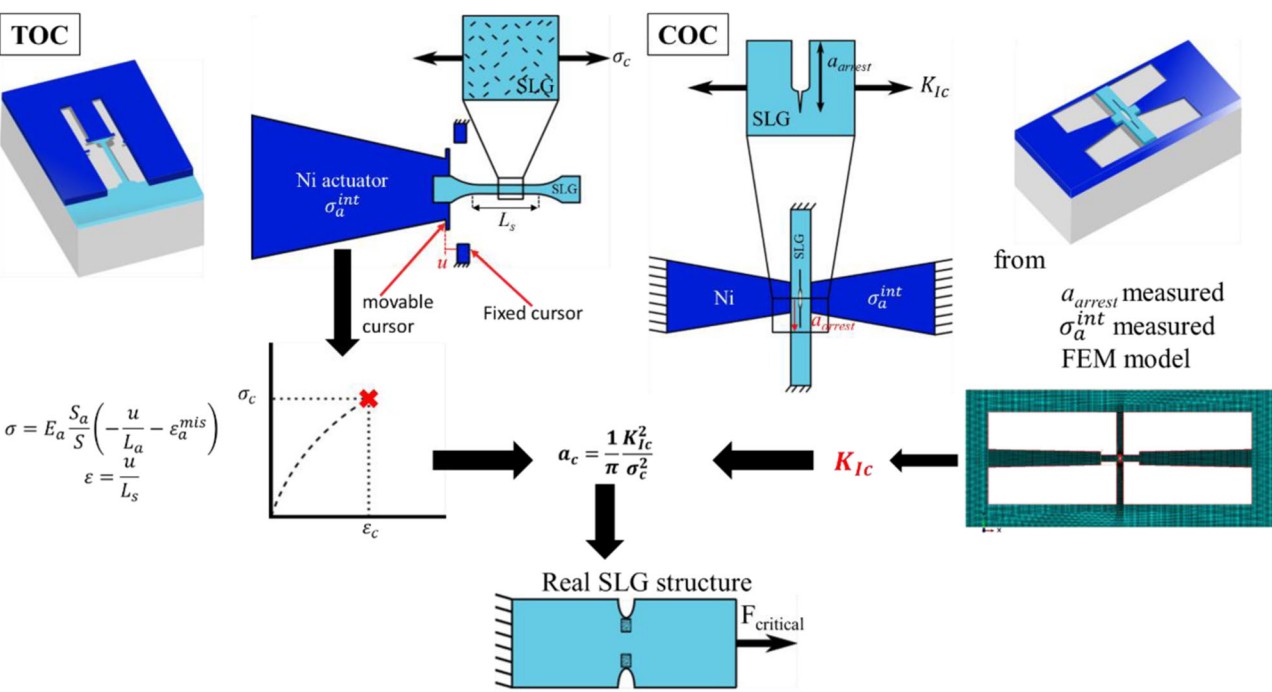

**Fig. 1 | Schematic of the on-chip mechanical testing set-up applied to single-layer graphene (SLG).** Both tensile-on-chip (TOC) and crack-on-chip (COC) configurations were combined to determine the critical flaw size that controls the failure of graphene membranes. The stress–strain response is given by TOC structures based on the displacement measured between movable and fixed cursors. The fracture toughness is determined using COC structures based on the measurement of the crack arrest length $a_{arrest}$ using a finite element model (FEM). The shaded blue sample represents SLG samples with defects, $a_c$ is the defect size that leads to fracture. The critical flaw size leading to failure $a_c$ can be estimated once the strength and fracture toughness are known from TOC and COC, respectively. The black arrows show that to determine ac, we need to use both data coming from TOC and COC. $E_a$ is the actuator Young's modulus, $S_a$ is the actuator area, $S$ is the specimen area, $u$ is the applied displacement, $L_a$ is the actuator length, $L_s$ is the specimen length, $\varepsilon_a^{mis}$ is the actuator mismatch, $\sigma_c$ is the strength of the studied material, $a_c$ is the critical flaw size, $\sigma_a^{int}$ is the residual stress of the actuator prior to release and $K_{Ic}$ is the fracture toughness.

case of the TOC configuration, the imposed displacement is generally applied to a dogbone specimen, while in the case of the COC configuration, the displacement is applied to a notched specimen to generate a crack from the tip of the notch. The working principle of the combination of the COC and TOC configurations into the new TOCOC test chain to determine the representative flaw size responsible for failure is illustrated in Fig. 1.

TOC principles have been applied to numerous thin-film systems, but never successfully adapted to 2D materials until now[14,28–33]. The specimen is deformed owing to the contraction of the attached actuator beam reaching a stable position corresponding to force equilibrium. The displacement is measured by SEM between movable and fixed cursors, as shown in Fig. 1. The internal elastic strain, also named mismatch strain in the actuator beam, is measured using dedicated structures[34,35]. The latter measurements, combined with a simple mechanical analysis[33], provide the stress-strain response (see Fig. 1 and Supplementary Note I) from which one can extract Young's modulus, strength, fracture strain, and strain hardening/softening behavior while being amenable to creep/relaxation measurements[32,36,37]. Each deformed specimen represents a single point on the stress-strain diagram. Thus, several TOC structures must be fabricated with different specimen and actuator beam lengths in order to vary the applied stress level covering the range from small elastic deformations up to fracture.

Recently, the COC concept has extended the potential of the on-chip approach, as inspired by Hatty et al.'s work[38], to extract the fracture toughness[39,40]. The core idea is that two actuator beams pull on a notched specimen. A crack initiates from the notch and then arrests (see Fig. 1). The fracture toughness is extracted from the final crack arrest length $a_{arrest}$, solving the problem of producing extremely sharp pre-cracks and associated artifacts, such as notch blunting effect at crack initiation. Benefiting from the lithography process to induce the pre-crack avoids the damage produced when using a focused ion beam. Finite element (FE) simulations, described in Supplementary Note II, are performed to determine $K_I$. Now, approximate analytical expressions have been derived as well to elucidate the effect of the different geometrical parameters and to guide the design of the test structures[39,40]. The magnitude of $K_I$ can be roughly estimated in most cases by the following expression (see Supplementary Note III for more details):

$$K_I = (1 - \nu_a)\sigma_a^{int}\sqrt{L_a}\frac{4\sqrt{\frac{6L_a}{\alpha_2 L_s}}}{32\frac{E_a}{E}\frac{a^2}{L_s^2} + \frac{L_s}{a}\frac{L_a}{W_a}\frac{t}{t_a}}, \qquad (1)$$

where $\nu_a$ is the Poisson ratio of the actuator, $\nu$ is the Poisson ratio of the test specimen (here graphene), $\alpha_2 = 1 - \nu^2$ in plane strain and $\alpha_2 = 1$ in plane stress, $\sigma_a^{int}$ is the residual stress in the actuator prior release, $L_a$ is the actuator length, $L_s$ is the specimen length, $E_a$ is Young's modulus of the actuator, $E$ is Young's modulus of the specimen, $t_a$ is the actuator thickness, $t$ is the specimen thickness, and $W_a$ is the actuator width. The fracture toughness $K_{Ic}$ can be estimated from Eq. (1) by using for the crack length $a$ the final crack arrest length $a_{arrest}$. But, once again, FE simulations have been systematically performed to extract more accurate values for the stress intensity factor.

Equation (1) indicates that the magnitude of the stress intensity factor $K_I$ is proportional to the residual stress in the actuator prior to release and to its length $L_a$, with a complex dependence on the crack length and on several other geometrical quantities. The parameters required to perform the FE simulations were determined experimentally, as given in Supplementary Table 1, with related uncertainties.

Here, we report the combined use of TOC and COC, TOCOC method, for the robust testing of 2D materials with dimensions representative of real applications, as demonstrated for monolayer CVD-graphene. In the TOCOC method, the strength of graphene $\sigma_c$

obtained from TOC structures is linked to the fracture toughness $K_{Ic}$ determined by COC structures based on $a_c = \frac{1}{\pi}\frac{K_{Ic}^2}{\sigma_c^2}$ which provides the critical defect size $a_c$ responsible for triggering the failure of graphene-based devices, as schematically explained in Fig. 1. These structures are fabricated following the steps schematically illustrated in Supplementary Fig. 1 and explained in the "Methods" section. High-quality monolayer CVD-graphene is investigated (see the "Methods" section and Supplementary Fig. 2).

## Results and discussion

### Fracture toughness of monolayer CVD-graphene

Figure 2 shows a set of successful COC structures with the crack arrested at some distance from the notch for asymmetric (shown in the panel of Fig. 2a, b) and symmetric (Fig. 2d, e) configurations. Out-of-plane deflection is limited, as highlighted in the zoom of Fig. 2c. In successful test structures, the crack follows a straight path. Now, many test structures turned out to be unsuccessful for several reasons described in Supplementary Note IV. Several dies were released, leading to 80 successful COC structures (58 asymmetric and 22 symmetric).

Figure 3a compares the cumulative probability distribution of $K_{Ic}$ for symmetric or asymmetric configurations. A majority of $K_{Ic}$ values from both configurations are in the same range. Each structure delivers a $K_{Ic}$ with a given uncertainty. An error propagation analysis was conducted as detailed in Supplementary Note V. The uncertainty on $K_{Ic}$ is ~17%, coming essentially from the uncertainty on the graphene Young's modulus. The FE simulations and the uncertainty study did not take into account graphene's anisotropic elastic behavior. As a matter of fact, the elastic modulus when determined for different in-plane loading directions using the elastic constants $C_{11}$ of 358 N/m and $C_{12}$ of 60 N/m (from ref. 41), involves <5% variation between maximum and minimum stiffness, which is within experimental uncertainty. Considering both geometries leads to a mean fracture toughness value of $4.4 \pm 0.1\,\text{MPa}\sqrt{\text{m}}$, close to the $4.0 \pm 0.6\,\text{MPa}\sqrt{\text{m}}$ found for BLG by Zhang et al.[19], but smaller compared to multilayer graphene (MLG) data[20,22]. The standard error of $0.1\,\text{MPa}\sqrt{\text{m}}$ is <17% since many specimens were analyzed. The theoretical mode I fracture toughness derived from analytical calculation[18] is $3.21–3.45\,\text{MPa}\sqrt{\text{m}}$, and from numerical (first principle) calculations[9] is $3.9 \pm 0.4\,\text{MPa}\sqrt{\text{m}}$, reasonably close to our results as well.

Figure 3b compares the different experimental values reported in the literature. Note that the maximum number of tests in the literature is seven tests by Hwangbo et al.[24], much smaller than the present 80 successful tests. The reported high $K_{Ic}$ values in the literature can be attributed to several factors. First, a multilayer can lead to toughening mechanisms due to the difference in the grain orientation in each layer and stacking order. Sliding at interfaces can also involve energy dissipation. Second, the initial crack is usually a notch and not a true precrack (~1 nm) in the sense of fracture mechanics, this is expected to nearly double the fracture toughness[42]. Third, the crack driving force can be reduced through crack branching[22,24]. The main reason behind the difference between the $K_{Ic}$ from asymmetric or symmetric structures is attributed to the out-of-plane displacement, as confirmed by recent FE simulations performed by Shafikov et al.[43]. In addition, graphene easily buckles/crumples due to its small bending modulus and its high in-plane stiffness as demonstrated by Euler's buckling theory[44]. This twisting becomes more dominant as the length increases, thus explaining the failure of most symmetric structures as will be discussed later.

Figure 3c summarizes all the reported fracture experiments on graphene as a function of the number of graphene layers and the area of the test specimen. The graphene membrane areas tested here are around 400 and ~800 μm² for the asymmetric or symmetric designs, respectively, while the specimens tested in the literature do not exceed a few μm².

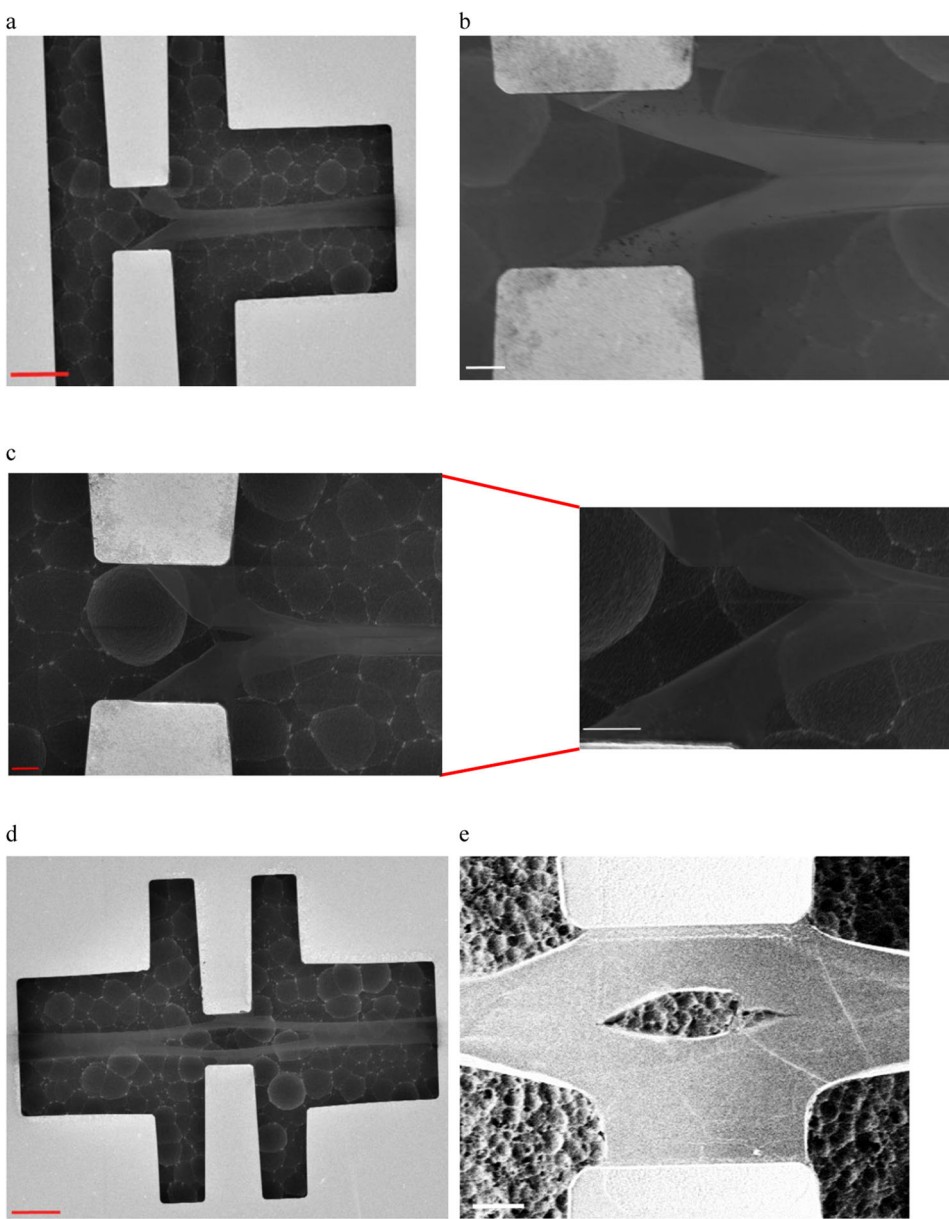

**Fig. 2 | Successful monolayer CVD-graphene COC structures. a** General view of an asymmetric structure in which a crack is initiated and propagated in a brittle way. **b** Closer view of an asymmetric structure shows a large crack tip opening displacement and the crack arrested after propagating over a significant distance. **c** Another asymmetric COC where the out-of-plane displacement is more evident, with a zoomed view (on the left side) showing the border between the notch and the crack. Strictly speaking, on the upper side of the notch, one can observe a kind of step marking the border between the notch and the initiated crack. **d** A general view of a successful symmetric configuration. **e** A zoomed view of the crack in a symmetric configuration where the crack opening is narrower on the left side compared with the right one- both crack lengths are close, and the cracking direction is similar. Red scale bar, 10 μm and white scale bar 2 μm.

A final point that deserves extra attention is how to explain the variations of $K_{Ic}$ from one specimen to another within the same test configuration. First, some values at the limit of the distribution are presumably associated with specimens exhibiting wrinkles near the notch tip, which are known to accelerate failure[17], although, in other studies, wrinkles are considered as offering an extra resistance to crack propagation[24]. In any case, wrinkles artificially modify the extracted value of the fracture toughness, an effect not accounted for in our uncertainty analysis. Although the transfer was performed in such a way as to avoid producing wrinkles in graphene, the removal of the underneath layer can also introduce wrinkles in graphene[45,46]. A wrinkled freestanding graphene tends to self-fold under deformation. This is indeed what has been observed for the widest specimens, as a way to release the in-plane strain energy. In this work, the presence of wrinkles near the crack tip is unlikely, especially in all the 80 measured specimens (see Supplementary Note VI for more details). For asymmetric design, creases have been sometimes observed and can be partially responsible for the (artificial) fracture toughness variation. Another possible artifact could be due to PMMA residues on graphene. Now, the bonding strength between PMMA residues and graphene is weak besides the fact that the $K_{Ic}$ of PMMA residues is low ~1 MPa$\sqrt{m}$[24]. Hence, PMMA residue islands can potentially affect the cracking rate and path but not the fracture toughness. Here, the cracking path was, in most cases, similar from one specimen to another excluding thus any significant impact of residues on $K_{Ic}$. Finally, a grain size effect on $K_{Ic}$ is expected when the grain size is extremely small, for instance,

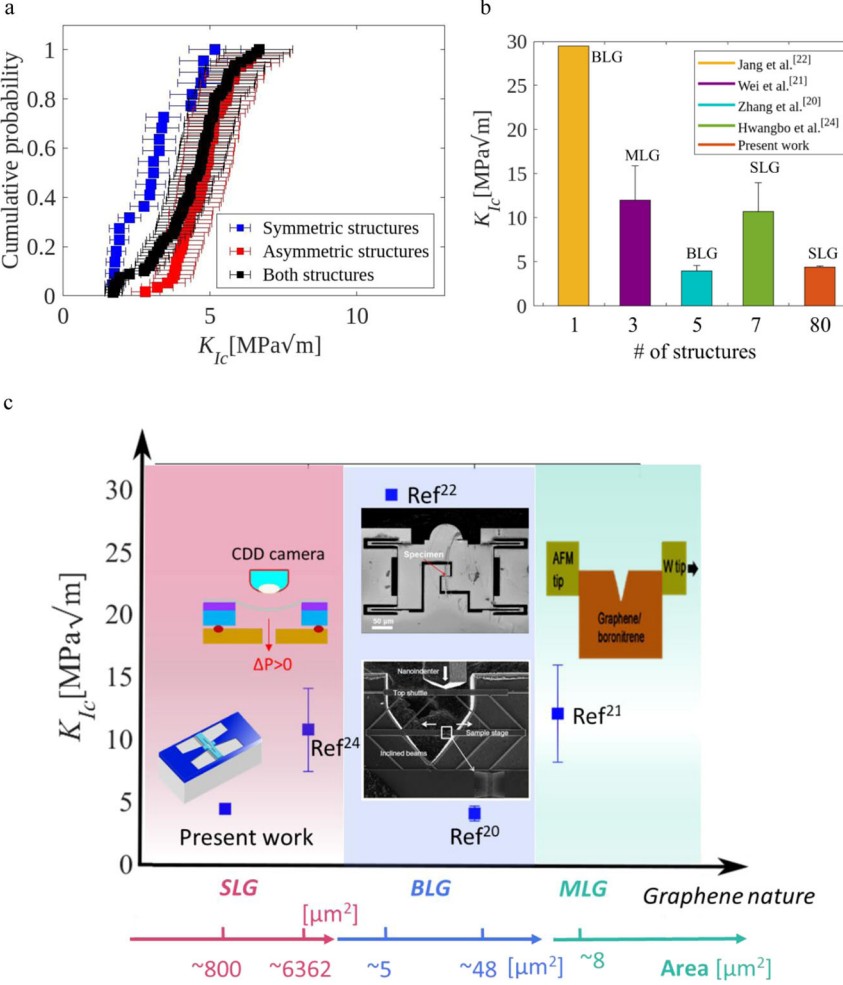

**Fig. 3 | Fracture toughness of graphene, from on-chip testing and literature. a** Fracture toughness distribution as a function of cumulative probability for symmetric structures only (in blue), for asymmetric structures only (in red), and both structures combined in black color. **b** Histograms comparing all performed experiments using graphene and the number of tests involved to determine the corresponding fracture toughness. (In addition, the nature of graphene is indicated on top of each bar. MLG is for multilayer graphene, BLG is for bilayer graphene, and SLG refers to single-layer graphene. The color code is shown in the upper right frame). **c** Comparison between the $K_{Ic}$ measured in this study, and the reported experimental values as a function of graphene layers and specimen area–the principle of each experimental technique is illustrated by a picture. The error bar in this work is too small, which is difficult to illustrate. The error bars refer to the standard deviation. "Source data are provided as a Source Data file".

around 250 Å[9]. However, here, the interaction of the crack tip with domain/grain boundaries can also be excluded since the graphene domains in which specimens have been patterned have a size of around 3 cm (see Supplementary Fig. 3). The dispersion is thus essentially attributed to the uncertainty on the measurement of the crack length as well as on minor warping-out-of-plane effects. In conclusion, the value $K_{Ic} = 4.4 \pm 0.1\,\mathrm{MPa}\sqrt{m}$ can be safely considered as the definitive fracture toughness for monocrystalline single-layer graphene.

**Strength and fracture strain of monolayer CVD-graphene**

Both standard TOC dogbone (Fig. 4a) and TOC rectangular specimens (Fig. 4b) were tested (more details in the "Methods" section). The motivation behind using rectangular graphene specimens was to reduce the tendency for graphene to crumpling/tubing, as often observed in this study for the dogbone geometry. Indeed, the wider sections combined with the constraints at the clamping extremities strongly reduce the propensity for topological changes under deformation. However, the regions near the clamped regions of the rectangular sample are not undergoing perfect uniaxial tension and constitute stress concentrators. Many successful TOC structures, i.e.

222 specimens, were produced with single-layer graphene, which, per se, is an experimental accomplishment.

Figure 5a, b shows the true stress-true strain curve for single-layer graphene specimens showing no twisting, warping or crumpling, or any other observable defects. In general, narrower specimens exhibit more twisting. Therefore, the data from the widest dogbone specimens are always close to perfect homogenous uniaxial tension. Furthermore, the widest specimens offer the largest resistance to the actuator beam, leading to a smaller uncertainty on the stress determination[30]. The results shown in Fig. 5 were obtained without considering any pre-tensile stress in the graphene. By considering pre-tensile stress levels of 0.05–0.6 N m$^{-1}$ [6,47,48,] the fracture strain and stress could change by about 1%, justifying the assumption of neglecting the graphene's pre-strain. Both configurations, as detailed in the "Methods" section, lead to Young's modulus of ~950 GPa, close to the expected value (1 TPa).

The discrepancy in the TOC results can be related to the presence of wrinkles. As mentioned earlier, the quantification of the impact of wrinkles on the extracted Young's modulus is a challenging problem that has been studied numerically. Shen et al.[49] showed a negligible impact of wrinkles on the stiffness of the monolayer graphene layer. In

a

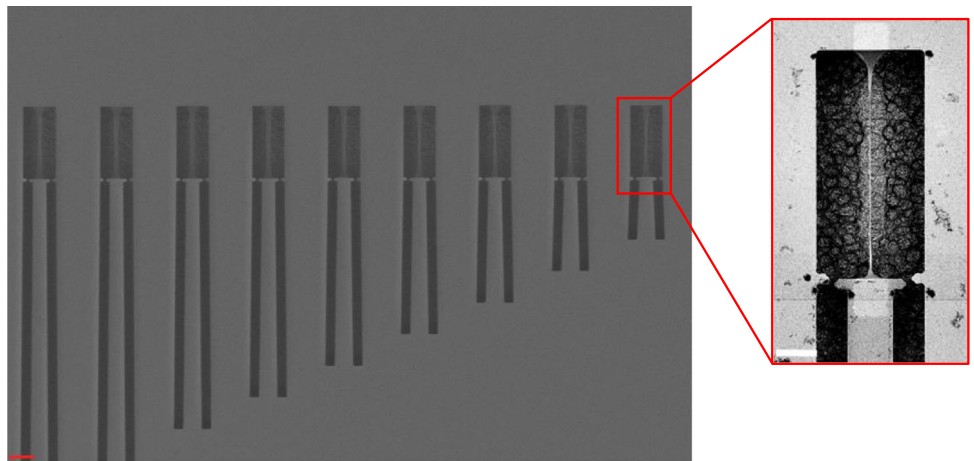

b

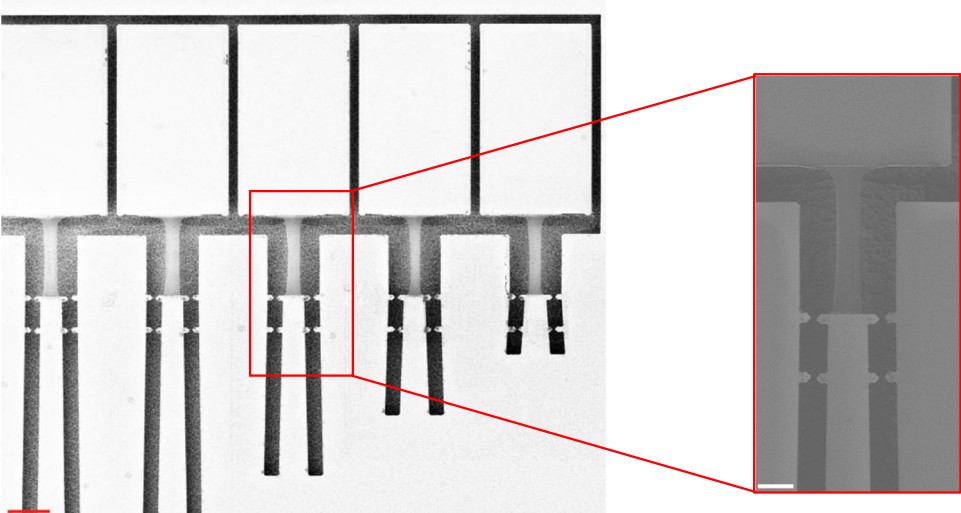

**Fig. 4 | On-chip uniaxial tensile testing of graphene. a** Series of 'dogbone' TOC structures with different actuator lengths. The inset is a zoomed SEM micrograph of a dogbone graphene specimen for the TOC test. **b** A series of 'Rect' TOC structures with a variety of actuator lengths. The inset is a zoomed view of a rectangular graphene specimen in the TOC configuration. The red scale bar refers to 20 μm and the white scale bar, is 10 μm.

the worst-case scenario, high wrinkle density with high wavelength involves an 11% reduction of Young's modulus along the armchair direction. About 4% reduction along a zigzag direction is in the range of the measurement error on the Young's modulus determined in this study. This value could be even lower since the wrinkle density in the present work is low. On the other hand, Qin et al.[50] reveal higher strength in the case of wrinkled graphene specimens compared with flat ones, which was used as an argument to strengthen some metal matrix composites[51]. Consequently, the extracted Young's modulus will not change drastically in the presence of a small wrinkle's amplitude. Another point worth mentioning is that the results obtained here are all consistent with one another, and when a test is repeated, even though corrugations are not uniformly distributed. This consolidates the trust in the validity of the results (more details are provided in Supplementary Note VI). Note finally that the main interest of the TOC structures was not to look at the elastic behavior but mainly to determine the fracture strain and corresponding fracture stress on a large set of specimens.

Figure 5a shows a maximum surviving strain of 0.045 with dogbone specimens, and 0.115 with rectangular ones (see Fig. 5b). This is the highest uniform deformation reached without failure in this study and, we believe, ever reported in the literature for truly uniaxial tension conditions. The maximum reported fracture strain[21] we found in the literature is ~5.8% for a ~12 μm² specimen area, which is half of the maximum value obtained here, and for a significantly larger specimen with a surface area equal to ~160 μm². The product of $1/2 \times$ fracture stress × fracture strain = $1/2 \times 110$ GPa $\times 0.115 \approx 6.3$ GJ m$^{-3}$ might be the largest density of mechanical energy ever stored in a freestanding material. The TOC working principle is such that the precision on the strain is much larger than on stress[30], as explained in Supplementary Note VII. Hence, the focus hereafter will be on the fracture strain, while the corresponding strength can be estimated by assuming linear elasticity and a 950 GPa Young's modulus.

Figure 5c is a plot of the deformation applied to all the test specimens as a function of the specimen length with a different symbol, whether the specimen is broken or not. Supplementary Fig. 4 shows

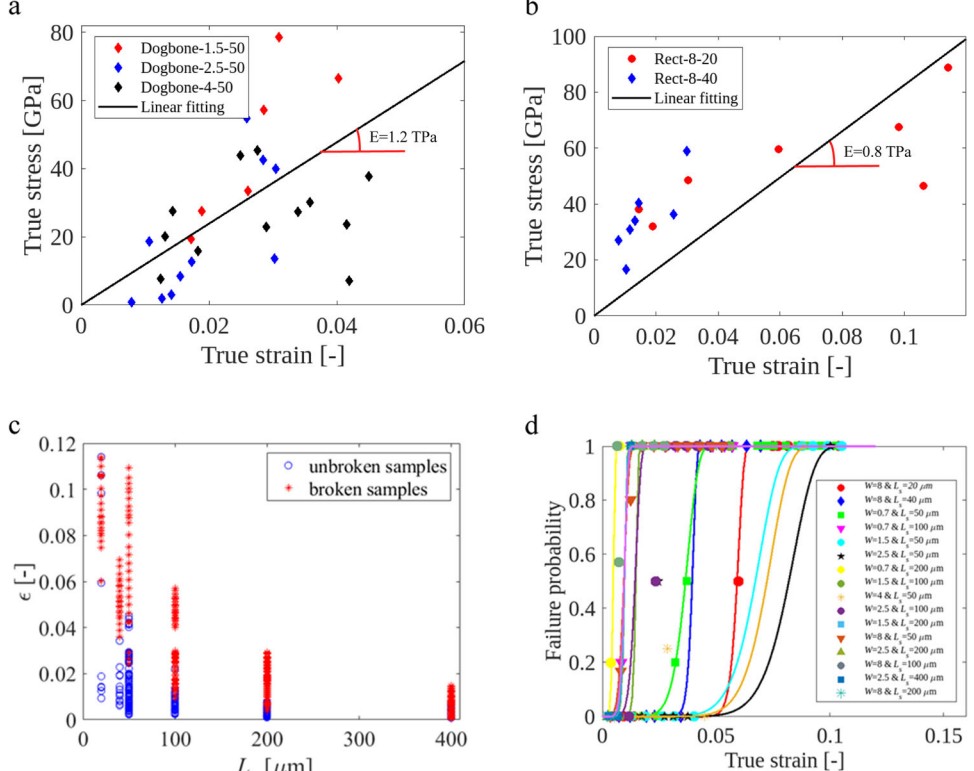

**Fig. 5 | Results from TOC structures on monolayer graphene. a** Uniaxial stress–strain response of single-layer graphene when considering only dogbone specimens, where 'Dogbone-$x$–$y$' refers to a specimen with a dogbone geometry, $x$ μm wide and $y$ μm long. **b** Stress–strain response of monolayer graphene obtained by considering only rectangular specimens, 'Rect-$x$–$y$' means rectangular sample where the first number is the width of the specimen and the second number is the specimen length. **c** Variation of the strain as a function of specimen length $L_s$ for both broken and unbroken specimens. The strain for an unbroken specimen is the

measured one. The strain corresponding to a broken specimen is the maximum strain applied to this specimen, known based on Eq. (S9). **d** Failure probability variation with true strain for different specimen lengths and widths, obtained from $N_{broken}^{\varepsilon}/(N_{broken}^{\varepsilon} + N_{unbroken}^{\varepsilon})$, where $N_{broken}^{\varepsilon}$ is the number of broken specimens undergoing a strain $\varepsilon$, while $N_{unbroken}^{\varepsilon}$ corresponds to unbroken specimens under strain $\varepsilon$. The solid line linking the data points is the fitted Weibull function, using Eq. (3). "Source data are provided as a Source Data file".

## Table 1 | Parameters of the Weibull distribution

| $L_s$ [μm] | 20 | 40 | 50 | | | | | 100 | | | | 200 | | | | 400 |
|---|---|---|---|---|---|---|---|---|---|---|---|---|---|---|---|---|
| $W$ [μm] | 8.0 | 8.0 | 0.7 | 1.5 | 2.5 | 4.0 | 8.0 | 0.7 | 1.5 | 2.5 | 8.0 | 0.7 | 1.5 | 2.5 | 8.0 | 2.5 |
| Weibull modulus, $m$ | 27 | 27 | 9 | 9 | 10 | 10 | 6 | 8 | 19 | 8 | 10 | 8 | 10 | 13 | 8 | 12 |
| $\varepsilon_0$ | 0.06 | 0.040 | 0.038 | 0.070 | 0.085 | 0.075 | 0.01 | 0.01 | 0.015 | 0.015 | 0.005 | 0.005 | 0.01 | 0.01 | 0.01 | 0.01 |
| Correlation coefficient, $R^2$ [%] | 83 | 100 | 99.9 | 100 | 75 | 93 | 97.3 | 100 | 100 | 99.5 | 79 | 96.6 | 100 | 100 | 100 | 100 |

$L_s$ is the specimen length and $\varepsilon_0$ is the characteristic strain.

the same data but as a function of the sample area. The determination of the failure probability of graphene TOC specimens is explained in the "Methods" section. Figure 5d plots the failure probability as a function of applied strain (taken as a mean value over a small interval) for different specimen lengths and widths for both rectangular and dogbone geometries, see the earlier application on aluminum films[31]. The solid lines in Fig. 5d represent the Weibull functions, with the parameters given in Table 1. The rectangular specimens are characterized by a high value of Weibull modulus $m = 27$, while the dogbone specimens exhibit a lower value of $m$, as detailed in Table 1. Lower $m$ are obtained for larger specimens indicating that another population of defects/imperfections is playing a role typically due to the twisting of the specimens. Hence, one can hardly rationalize these data into one single master plot[52].

Pugno and Ruoff[53] already applied Weibull statistics at the nanoscale to carbon nanotubes, noting that point defects more than

length or area/volume defects were predominant, as similarly observed here and in contrast to classical (volume-based) Weibull's statistics. In the work of Cui et al.[54], a value of $m = 13.9$ was determined experimentally under static loading, and lower values of $m$ equal to 6.4 and 4.3 were determined under cyclic loading of $10^9$ and $10^7$ cycles respectively, while Shekhawat et al.[9] performed MD simulations that led to $m = 10.7$. Our values for dogbone specimens are close to the latter reported $m$ values. The variation of $m$, particularly between the dogbone and rectangular graphene samples, is mainly attributed to the size and geometry differences and not to the material property since $m$ is the same for a given material. For instance, wider samples tend to remain more in-plane compared with the longer and narrower samples that likely folded acting like nanotubes more than a flat sheet. In another vein, the variation of Weibull moduli in this work reflects the sensitivity of CVD-monolayer graphene to the presence of defects such as creases that can deteriorate under cyclic loading, as confirmed by

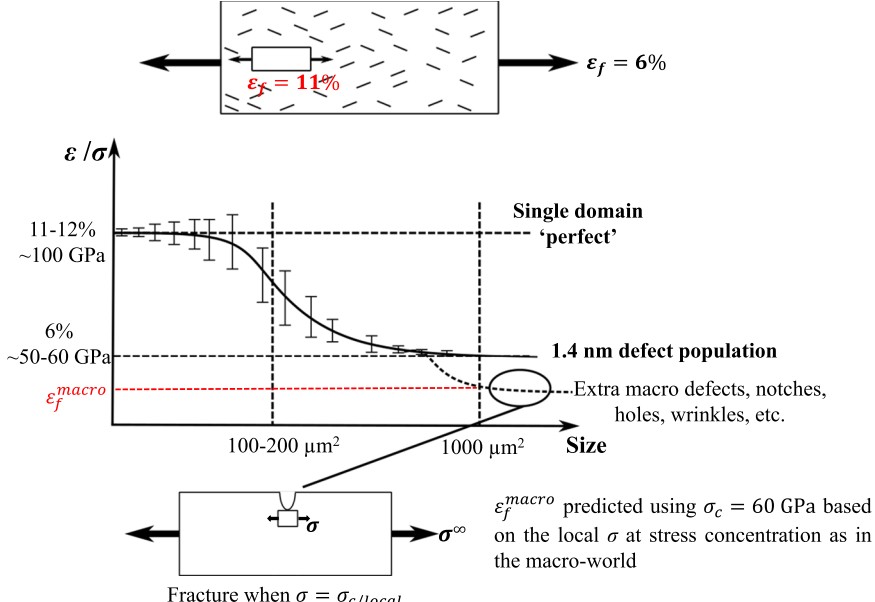

**Fig. 6 | Schematic representation of how the fracture strain and strength vary as a function of the population of defects and specimen size.** Extremely small specimens can sometimes be immune to any internal flaw and reach up to 11% fracture strain. Nevertheless, in general, intrinsic flaws are unavoidable for large micro-size specimens (and larger), and the fracture strain reaches the value of 6%. Lower fracture strain can be found if another population of flaws is generated or in the case of additional stress concentrators.

Cui et al.[54]. One more note, the high value of $m$ indicates a high quality of graphene, especially for rectangular samples.

In the cleanest tensile tests performed in this work with limited "macroscopic" stress concentration and twisting, i.e. with the short $L_s = 20$ μm dogbone specimens, no specimen failed below 6%. The corresponding strength is equal to 57 GPa (using $E = 950$ GPa). Based on the fracture mechanics equation $K_{Ic} = \sigma_c \sqrt{\pi a_c}$ with $\sigma_c$ the overall fracture stress and $K_{Ic} = 4.4 \pm 0.1$ MPa$\sqrt{m}$, one can estimate the typical flaw size according to fracture mechanics, i.e. $a_c \cong 1.90$ nm. Fracture mechanics cannot predict ideal strength and cannot treat small-scale as well as not perfectly sharp defects, for this reason, quantized fracture mechanics (QFM) was introduced in Pugno and Ruoff[18]. According to QFM, the prediction of the crack size is not $a_c$ but instead is $a_c' = a_c - b/2$ where $b$ is the fracture quantum (for graphene it is considered to be nearly the distance between adjacent atoms, namely $b = 0.25$ nm, but expected to increase with the size-scale itself). For CVD graphene assuming a process temperature of 1000 or 1500 K, a thermodynamic[55] (lower bound) vacancy fraction of $5.66 \times 10^{-36}$ or $3.18 \times 10^{-24}$ was estimated, respectively, thus resulting in the specimens tested here (1000 μm² involves ~$10^{10}$ atoms) in potentially defect-free structures. Assuming this scenario ($a_c' = 0$) for the 110 GPa specimens results in $b \cong 1.0$ nm (considering the measured fracture toughness), and, accordingly, $a_c' \cong 1.4$ nm for the ~57 GPa strength.

The fracture strain determined from modeling in the case of armchair edge[10,12,56] (~13%) is also not far from the largest experimental fracture strain from TOC specimens (~11.5%). However, the lack of studies investigating the edge effect on mechanical properties of monolayer graphene does not alter the fact that the obtained results in the case of graphene nanoribbons could be valid for graphene too. The effect of free edge warping in graphene nanoribbons was investigated numerically, revealing a decrease of Young's modulus[57] hence smaller strength, which can be valid for SLG. Thus, if the edge effect was taken into account during simulation, it probably resulted in a smaller Young's modulus, hence a smaller strength value. On a different note, the fact that larger specimens with much lower fracture strain are sometimes detected is not necessarily due to a wide distribution of flaw sizes, otherwise, also smaller specimens would fail sometimes at a strain below 6%, but to other extrinsic effects associated with experimental artifacts (specimen preparation, twisting/warping, etc.).

Accordingly, the key question that arises from these results is the origin and nature of these 1.4 nm-long defects. The most likely origin is the pentagon–heptagon pair defects. Their length, based on the work of Terdalkar et al.[11] is equal to 1.7–1.9 nm. Seemingly, the presence of these 5–7 defects as well as Stone–Thrower–Wales have been demonstrated to reduce drastically the strength and the fracture strain[58]. Under stress, a 5–7 defect leads to an elliptical cavity with a size between 1.5 and 2 nm close to the 1.4 nm defect derived in our analysis, which could act as a starter crack, as anticipated from fracture mechanics analysis.

Figure 6 shows the huge impact of 1.4 nm defects on the strength and fracture strain, especially when the specimen size increases, indicating that it is "almost impossible to fully get rid of these defects", as originally discussed in[18], except for extremely small (sub-micron) pristine specimens.

The strength is sensitive to specimen size, defects, as well as grain size. The Weibull analysis, as performed here, is the conventional approach used in the macroscopic world to design fail-safe structures and devices, which must also be used for the fail-safe design of graphene-based elements.

In summary, the mechanical properties of freestanding CVD-monolayer graphene were investigated using a new tension and crack on-chip (TOCOC) combination. The method relies on the use of tensile residual stress in an actuator beam that, once released, pulls on a notched graphene specimen to extract the fracture toughness or on a uniform tensile specimen to determine the uniaxial stress–strain response. The value $K_{Ic} = 4.4 \pm 0.1$ MPa$\sqrt{m}$ is established as the definitive fracture toughness of single-layer graphene based on 80 successful experimental specimens. This fracture toughness value is higher than any hard thin layer in the sub-100 nm range, although still very low when compared with bulk tough steels or high entropy alloys with values above 250 MPa$\sqrt{m}$ because energy dissipation at the crack tip is limited by the thickness. The largest experimentally measured strain ever of 11.5% was determined over a large specimen area 160 μm². Furthermore, Young's modulus of 0.95 TPa and the

maximum strength of 110 GPa were found, close to the theoretical value and corresponding to defect-free graphene, as also confirmed by thermodynamics and quantized fracture mechanics. The mean strength is smaller than the theoretical value, being often controlled by the presence of 1.4 nm defects associated with 5–7 pairs. Such defects are most probably unavoidable with upscaled processing methods and will always lead to a maximum design strain of 5–6%, corresponding to a fracture stress of 50–60 GPa. This value should be considered as the best strength to be used for the design of graphene-based structures, except if less ideal processing leads to an additional population of larger defects, which would then result in lower strength. This simple and robust approach can be applied to other 2D materials for answering several open scientific questions related to mechanical behavior in current literature.

## Methods

### Fabrication process of COC and TOC

Some of the fabrication steps followed for designing graphene-based on-chip test structures differ from earlier studies on thicker films[14,28–37]. First, alignment marks are patterned, see Supplementary Fig. 1II–III, followed by the deposition of a thin gold layer as shown in Supplementary Fig. 1IV. After lift-off, the CVD-graphene is transferred onto the Si substrate (Supplementary Fig. 1V). The synthesis of graphene and transfer techniques are explained in the next subsections. Positive lithography (Supplementary Fig. 1VII) is performed to pattern the graphene specimen. PMMA is coated before depositing a positive photoresist to protect graphene (Supplementary Fig. 1VI). The unprotected parts of graphene after development are etched using oxygen plasma, and the remaining resist is removed by warm acetone (Supplementary Fig. 1VIII). The second lithography is carried out to pattern the actuator layer (Supplementary Fig. 1IX, X). This is followed by electron beam deposition of a nickel layer (Supplementary Fig. 1XI), which contains high tensile internal stress of around 600 MPa.

A 5-nm-thick Cr layer was deposited before the Ni layer to enhance the adhesion between graphene and Ni as well as between Si and Ni. The lift-off is done using hot acetone for a long duration to get rid of the resist residues (Supplementary Fig. 1XII). For TOC, one actuator beam is used, as shown in Supplementary Fig. 1XVI, while for COC, two actuator beams are designed as displayed in dark blue color in Supplementary Fig. 1XIV, XV, for asymmetric and symmetric configurations, respectively. By etching the Si substrate surface using $XeF_2$ (Supplementary Fig. 1XIII), the actuator beam is released from the substrate, and contracts, acting as a spring to impose a displacement on the test specimen. For the TOC method, the displacement is measured between two cursors, as highlighted in the inset of Supplementary Fig. 1XVI. While, for the COC, the measured parameter after release is the crack arrest length (Supplementary Fig. 1XV).

### Graphene growth

Graphene is synthesized on copper foil (50 μm-thick, oxygen-free high conductivity, 99% purity, advent research materials) in a low-pressure chemical vapor deposition (LPCVD) system. The reactor is a horizontal hot-wall quartz tube. The gas inlet is monitored by 3 mass flow controllers connected to a pure Ar (Praxair, Research Grade, 99.9999%), an $H_2$/Ar (10%), and a $CH_4$/Ar (2000 ppm) gas bottle. The copper foil has been rinsed with diluted HCl and deionized water to eliminate possible surface contamination before being loaded into the CVD tube. The CVD tube is heated up from 22 to 1050 °C for 60 min, and the copper foil is annealed in pure Ar at a pressure of 800 mbar for 60 min. After the annealing, 20 sccm $H_2$/Ar is injected into the tube to remove the copper oxide on the surface of the copper foil at a pressure of 220 mbar for 10 min. 32 sccm of $CH_4$/Ar and 320 sccm of $H_2$/Ar are injected into the tube to start the growth of graphene at a pressure of 180 mbar. The $CH_4$/Ar and $H_2$/Ar ratios have been maintained during the cooling down of the tube. The separated graphene domain has

been generated during the growth time of 90 min, as shown in Supplementary Fig. 5. When the growth time increases, the graphene domains merge together leading thus to a continuous graphene film. SEM and Raman's characterization confirmed that graphene is a monolayer, as shown in Supplementary Fig. 2.

### Graphene transfer method

A PMMA-assisted method is used to transfer graphene. PMMA (950 PMMA A9, MircroChem) diluted in anisole (99%, Sigma-Aldrich) is directly spin-coated on graphene for one minute. PMMA has been dried in an ambient atmosphere for 24 h and the graphene on the backside of the copper foil has been etched away by Oxygen Plasma. $FeCl_3$ is used to etch the copper substrate, and the PMMA/graphene stack has been carefully rinsed with deionized water more than 5 times to remove the metallic contamination as much as possible. Then, the PMMA/graphene stack is transferred to the target substrate and dried in an ambient atmosphere for 24 h in order to remove the residue water between the graphene and the target substrate. The specimen is baked in an oven for 15 min at 150 °C. At the end, the PMMA is removed using a warm acetone bath (50 °C) for 1 h.

### Description of the TOC method

Dogbone specimens have been produced with the following lengths: 50, 100, 200, and 400 μm. In addition, many widths were tested equal to 1, 1.5, 2.5, 4, and 8 μm. For rectangular specimens, a single width equal to 8 μm was used; while two different lengths were tested, equal to 20 and 40 μm. Here, the rectangular specimen will be defined as 'Rect'. The dogbone is simply noted as 'Dogbone'. The actuator length also varies to apply different levels of force, as shown in Fig. 4. This variation is between 30 and 1500 μm with a step of 30 μm for 'Rect' structures and between 50 and 1500 μm with a step of 25 μm for the 'Dogbone' configuration. The design of different specimen lengths is needed to characterize both large and small deformation regimes, using short and long specimens, respectively. These changes in specimen length, along with the different load levels allowed by varying the actuator length, are required in the case of materials exhibiting large fracture strains such as graphene.

### Failure probability

The failure stress is never known exactly with the TOC method because a specimen is either broken (and one does not know exactly the strain at which it was broken) or unbroken, as there is no continuous monitoring of the specimen during the deformation process (which occurs progressively thanks to the tapering of the actuator beams until the release of the last attached point, which can be considered as a fast release). Hence, one can generate, as a function of the applied stress or strain (or as a function of a small range of applied stress or strain), a failure probability $P_f$ by counting the number of broken specimens compared with the total number. This analysis can be repeated for different specimen sizes. The basic Weibull[15] analysis assumes that the failure probability is given by

$$P_f = 1 - exp\left[-\left(\frac{\sigma_N}{\sigma_0}\right)^m\right] \qquad (2)$$

with Weibull exponent $m$, characteristic strength $\sigma_0$, and nominal strength $\sigma_N$. The equation is identical when expressed in terms of the applied strain $\varepsilon$ and characteristic strain $\varepsilon_0$ assuming a linear elastic response. The justification of the exponential form is grounded on a weakest link model statistics, and thanks to the simplicity of the exponential functions, it satisfies the two limits, $P_f = 0$ when $\sigma_N = 0$ and $P_f \rightarrow 1$ when $\sigma_N \rightarrow +\infty$. The strength $\sigma_0$ depends on the specimen dimensions and corresponds to the stress with a probability of failure $P_f = 1 - e^{-1} = 0.63$. In the present work, the accuracy of the strain measurement is higher than the stress measurement, as explained in

Supplementary Note VII thus, using the Weibull equation as a function of the strain seems more convenient

$$P_f = 1 - exp\left[-\left(\frac{\varepsilon_N}{\varepsilon_0}\right)^m\right] \qquad (3)$$

Equations (2) and (3) can be used with other reference parameters, the key information being in the Weibull exponent $m$.

### Raman's characterization of graphene

Raman spectroscopy measurements were performed at three points (A–C), as shown in Supplementary Fig. 2a. In these three points, the 2D-band peak is sharp and narrow, as observed in Supplementary Fig. 2b. The intensity of the ratio G-band/2D-band peak is equal to 0.5, indicating that the graphene is a single layer. Moreover, graphene is of high quality since the D peak is negligible.

## Data availability

Source data are provided with this paper. Any additional requests for information can be directed to, and will be fulfilled by, the corresponding authors. Source data are provided with this paper.

## Code availability

The Python code used to calculate $K_I$ at crack arrest (and thus $K_{Ic}$) is made available as a supplementary file.

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

## Acknowledgements

The authors would like to warmly thank the support of the UCLouvain WINFAB cleanroom team. This work was supported by the ARC project Naturist (Convention No. 11/16-037) and by the FNRS under Grant PDR-T.0178.19. S.J. acknowledges the support assistant mandate from UCLouvain. N.M.P. gratefully acknowledge funding by the European Comission through the GrapheneCore3 881603 Project.

## Author contributions

S. Jaddi wrote and edited the original manuscript, designed and fabricated the used on-chip tensile and crack on-chip devices, carried out most of the experiments, measurements, and simulations and analyzed the results. M.W. Malik conducted graphene growth, transfer, characterization, and Raman analysis. B. Wang contributed to the graphene growth, transfer, and cleaning of the samples. N.M. Pugno contributed to data analysis and reviewing the paper. Y. Zeng supervised this work. M. Coulombier assisted in designing the samples and results discussion. J.-P. Raskin contributed to reviewing the paper, results analysis and supervising the present research. T. Pardoen wrote, reviewed the manuscript, assisted in discussing the results and supervised this research project.

## Competing interests

The authors declare no competing interests.
