## [Peer Review File · Nature Communications]

REVIEWER COMMENTS

Reviewer #1 (Remarks to the Author):

The manuscript entitled Definitive engineering strength and fracture toughness of graphene through on-chip nanomechanics presents a detailed study of the strength and fracture toughness of graphene that includes a large sample set of approximately 80 samples. This study should be of significant interest to the Nature Communication readership as it present one of the first reports of a large data set of mechanical properties suitable for fail-safe engineering design using graphene. The study is well presented and the conclusions appear to be sound. The following comments may be considered by the authors prior to publication.

- Can the authors comment on the variability on the dogbone shapes of graphene samples tested? As well as wrinkles in the graphene samples? Does any shape irregularities or wrinkles have any influence on the mechanical behaviour?
- How does the fail-safe fracture toughness reported compare to other relevant engineering materials?
- Does any residue remain on the graphene after transfer? How is it ensured that this is removed? Does any residue influence the mechanical measurements?

Reviewer #2 (Remarks to the Author):

In this manuscript, the authors report a very comprehensive experimental effort to measure fracture strength and toughness of graphene, through a microfabricated platform that allows uniaxial and notched-specimen testing of many samples. The authors report 80 results for the toughness of graphene, and about ~50 (this reviewer's estimation) data for strength. As such, this paper represents an unprecedented effort to characterize these quantities with statistical significance. The experiments themselves are a praiseworthy achievement. The toughness reported is generally in line with previous reports, although the amount of data gives greater confidence to this result. The maximum strength reported also falls in line with expectations, but the variation with size is nicely shown experimentally.

I recommend publication of this manuscript, after the following major revisions are addressed:

1. The authors should explain better two aspects of the notched test methodology: i) Although many details of the fabrication are given, I could not see explicitly stated how the initial notch is fabricated. Is it

lithographically patterned? I understand a sharp crack emanates from this notch, which later arrests itself at a given length. ii) Please add a succinct explanation of the K_{Ic} calculation in the main text. It is explained in detail in the supplementary material, but a short explanation with the main gist should be given in the main.

2. Please explain the influence of creasing in the tensile testing results. A thin film stretched as in the dogbone or rectangular samples will unavoidably develop longitudinal wrinkles or creases. I expected the finite-element simulations or theoretical model to address this, but it is not included. It is tangentially mentioned in the manuscript as "twisting". The authors should show this effect does not significantly influence the results. Creasing can lead to observed stiffening, which is a structural effect, and not an intrinsic property of the material.

3. The grain boundary size is presented somewhat ambiguously. The authors seem to suggest they test mono-crystal graphene in most cases, because the grain size is $\sim 3\text{cm}$, supported by Figure S3. This figure is a photograph of graphene on copper, so I am puzzled why this picture demonstrates large grains. Furthermore, Figure S2 shows a grain size of $\sim 200\mu\text{m}$. The toughness results probably are not affected by this (sample size $\sim 10\mu\text{m}$) but the large tensile samples may have some boundaries. This is not necessarily a problem. These experiments are difficult. But if boundaries are potentially present, this should be acknowledged.

4. Optional. The Weibull analysis could be refined. The authors are reporting different m exponents for different sizes. Bazant and others have shown that a given material has the same m , and the size is what causes the variation in the distributions. In other words, strictly speaking, using different m 's for the same material is incorrect, although it can give a workable description of the results. The authors could consider using the methods from Mechanics of Materials 162, 104057, 2021, which can allow the calculation of parameters from the strength data that could be compared with defect size and theoretical strength, to extract further insights from their data.

Reviewer #3 (Remarks to the Author):

This paper reports an on-chip method used to measure fracture toughness, stress, and strain of monolayer graphene. Residual stresses in deposited films are used to mechanically load graphene, and the final strain is measured by SEM. Many samples are measured to provide high statistical confidence in the fracture properties. Analysis of the results led to the insightful conclusion that 5-7 defect structures, having 1.4 nm length and prevalent in even high-quality graphene samples, are the limiting factor in graphene's fracture strength.

The experiments reported are clever, challenging, and provide greatly improved bounds on fracture properties of graphene. The statistical analysis is well explained and the discussion provides thoughtful insights. I provide several minor comments below, but overall find the work to be of high quality and interest. I recommend it for publication after minor revision.

1. Work on suspended graphene drums has reported pre-strain of the graphene when transferred/pinned to a solid wafer. Can you comment on whether pre-strain may be present in your graphene films and what effect this would have on the measured fracture strain etc?
2. The release/deformation process is said to occur fast (L298). Is it possible that there will be overshoot in this process, providing higher peak stress and strain than recorded in the SEM in the final settled state?
3. While most PMMA is removed from graphene by acetone, residue usually remains. Will this have any effect on your measurements?
4. The finite element analysis and the stress intensity equations appear to rely on the assumption of an isotropic continuum material. However, the measurements are of monocrystal, monolayer graphene. Can you please comment on the validity or any error introduced in these calculations as a result?
5. I suggest providing more of an explanation of the chip structure and how the chip test works in the main paper. Although the methods and supplementary provide more detail, it would be easier to follow with a little more explanation up front. Lines 81-95 reference fig1 a few times for details, including the cursors and how a crack propagates, that are not clearly labelled. FigS1 provides more detail, but both figs would benefit from more words in the caption, labels in the panels, and text describing the fig/procedure.
6. I find the conclusion that 5-7 defects limit the fracture strength of graphene to be quite insightful and supported by the analysis. The point made that small enough areas of graphene could possibly be free from such defects and survive larger strains is also interesting. However, I find that the way this is referenced in the abstract unclear before reading the full paper ("Except for micron-sized membranes, which could be designed based on 110 GPa strength, fail-safe design should be based on 57 GPa strength"). I suggest revising this sentence of the abstract to make the point clearer to anyone who only reads that far.
7. Some variables in fig1/ table1 seem to not be defined until later in the paper. Would be good to define in text or captions.

8. The reference numbers in Fig2b seem to be wrong.

9. Please state in the caption what the error bars in fig2b show. It appears to be standard deviation or something similar because the bar with only one sample has no error bars. I suggest also noting that the error bars for this one are not shown/calculated because there was one sample, or otherwise estimating the measurement uncertainty.

10. Fig5a&b the numbers in the legend are not explained until the methods section. I suggest explaining them in the main text or caption.

Responses to the Reviewers concerning the manuscript

NCOMMS-23-25673

Definitive engineering strength and fracture toughness of graphene through on-chip nanomechanics

The authors would like to sincerely thank the three Referees for the time spent in analyzing our manuscript and research findings as well as for the constructive suggestions and remarks. We are of course very pleased by the fact that our work was positively appreciated. All comments were taken into account very seriously and led to several changes in the revised manuscript, admitting that one or two of these comments were found quite challenging to fully satisfy without launching a long-term research effort. We provide hereafter a detailed set of answers to the comments and explanations about the changes made in the manuscript.

Referee 1

General comment. The manuscript entitled Definitive engineering strength and fracture toughness of graphene through on-chip nanomechanics presents a detailed study of the strength and fracture toughness of graphene that includes a large sample set of approximately 80 samples. This study should be of significant interest to the Nature Communication readership as it presents one of the first reports of a large data set of mechanical properties suitable for fail-safe engineering design using graphene. The study is well presented and the conclusions appear to be sound. The following comments may be considered by the authors prior to publication.

Thank you again for this positive opinion of the study.

Comment 1. Can the authors comment on the variability on the dogbone shapes of graphene samples tested? As well as wrinkles in the graphene samples? Does any shape irregularities or wrinkles have any influence on the mechanical behaviour?

Thank you for this question which also attracted our attention in the course of the research. We can (we think) convincingly answer the first part of the question but the second part is more challenging, see next.

The so-called dogbone shape is the usual geometry that is used to test material specimens under conditions that are as close as possible to uniaxial tension inside the reduced gauge section. Different dogbone lengths were designed in order to allow capturing the small as well as the large deformation regimes, with the longest and smallest specimens, respectively. As a matter of fact, for brittle materials, typically failing below or around 1% strain, a single specimen length (and variable actuator lengths) does the job, but for ductile materials (and graphene turns

out to have large fracture strains similar to some ductile films), one must play with different specimen lengths. The fact that the dogbone specimens include a reduction of the section makes the gauge section very narrow, and slight asymmetries and misalignments associated with the lithography of the dogbone shape easily lead to partial warping/tubing of the specimens. This tubing effect is observed on several specimens, and in the end, makes the dogbone geometry less perfect in terms of loading conditions than the basic rectangular beams (although, in general, for thicker conventional materials, it is the opposite). In the rectangular specimens, the constraint induced by the clamping extremities limits very much the sensitivity to crumpling/tubing at the expense of having to face the Saint Venant principle with small regions near the anchor points not undergoing perfect uniaxial tension (and leading to some stress concentration).

This aspect has been more carefully addressed in the revised manuscript:

“The motivation behind using rectangular graphene specimens was to reduce the tendency for graphene to crumpling/tubing as often observed in the case of the dogbone geometry. Indeed, the wider sections combined with the constraints at the clamping extremities strongly reduce the propensity for topological changes under deformation. However, the regions near the clamped region of the rectangular samples are not undergoing perfect uniaxial tension and constitute stress concentrators.”

More details in the Methods section:

“The design of different specimen lengths is needed to characterize both large and small deformation regimes, using short and long specimens, respectively. These changes in specimen length along with the different load levels allowed by varying the actuator length are essential in the case of materials exhibiting large fracture strains such as graphene. “

Now, we address the comments regarding the wrinkling. The presence of corrugations in graphene (wrinkles, ripples, or crumples) in 2D materials, unlike in 0D or 1D counterparts, can be attributed to numerous factors such as dislocations, surface anchorage, substrate relaxation, edge or interatomic interactions instabilities, and surface tension caused by solvent. Wrinkles/ripples are dependent mainly on the surface morphology of the growth substrate and the transfer process. They can hardly be suppressed, except with very specific fine-tuned methods, e.g. Chatterjee *et al.*, Chem Mater 2020.

Moreover, indeed, despite the great care taken during the graphene transfer step, we indeed still see some wrinkles in our specimens. Nevertheless, the graphene we have been testing has a low density of wrinkles thanks to the transfer method based on soaking PMMA/graphene layer in deionized water at a temperature of around 80°C and making sure the Si surface is hydrophobic. The advantage of this transfer technique was reported in several works, e.g., N. Liu, *et al.* (Nano Res 2011) and L. Gao, *et al.* (Nature 2014). Moreover, and this is an important argument to answer the Referee’s comment, the region in which the test structures were patterned, was selected carefully in a zone with a minimum amount of wrinkles.

However, although as said above, we minimized the presence of wrinkles when designing the test specimens, the etching of the underlayer unfortunately re-introduces wrinkles into graphene that becomes unstable, tending to self-fold especially with wide samples as observed in this work, in Lambin *et al.* (*Appl. Sci.* 2014) and Cranford *et al.* (*Appl. Phys. Lett.* 2009). These out-of-plane ripples/wrinkles manifest in freestanding graphene as a way to release the in-plane

strain energy. The wrinkle wavelength and density increase under an externally applied deformation.

Considering the explanation above, we confirm that our graphene specimens do sometimes contain wrinkles and ripples and we cannot think of a method to completely get rid of these corrugations. Hence, we are left with the question of the impact on the measured properties. The influence of corrugations on the extracted mechanical properties is not a trivial question. It is very challenging to quantify, and up to now, to our knowledge, there is no work studying this impact experimentally in detail. Nevertheless, some studies based on numerical models and theoretical calculations were conducted to investigate the potential effects. The work of Xi Shen *et al.* (Carbon 2014) shows the negligible impact of wrinkles on the mechanical response of the monolayer graphene layer. In the worst-case scenario, a high wrinkle density with a high wavelength shows around 11% reduction of Young's modulus along the armchair direction. Around 4% reduction is found along the zigzag direction which is in the range of the measurement error on the Young's modulus determined in our study (0.85 TPa and 1.2 TPa resulted from dogbone and rectangular samples, respectively). On the other hand, some studies showed larger strength for wrinkled graphene compared to flat graphene like the work of Qin *et al.* published in Nanoscale in 2017. In the same work, a negative Poisson ratio was demonstrated. Due to this high strength, wrinkled graphene was used to strengthen some metal matrix composites as in Zhao *et al.*, Carbon 2020 and in J. Mater. Sci. Technol. 2022. It has been reported by Akhunova *et al.* work's (Appl. Sci. 2022) that small-amplitude corrugations in graphene do not reduce drastically its Young's modulus and fracture strength contrarily to high-amplitude corrugations. The graphene samples tested in the present work exhibit only small-amplitude corrugations that can be expected to slightly reduce the estimated Young's modulus and fracture strength compared with the true values.

What can be added on our side, to consolidate the trust in the results, is that the Young's modulus that is extracted (950 GPa) is the expected one and that the results are all consistent with one another when data are generated multiple times, even though wrinkles will not be uniformly distributed and will vary from one specimen to another.

We have included some comments regarding this aspect in when discussing the COC and TOC results in the revised version to recognize more clearly the presence of wrinkles and the key comments above :

For COC results,

'First, some values at the limit of the distribution are presumably associated with specimens exhibiting wrinkles near the notch tip, which are known to accelerate failure,^{Erreur ! Source du renvoi introuvable.} although, in other studies, wrinkles are considered as offering an extra resistance to crack propagation.^{Erreur ! Source du renvoi introuvable.} In any case, wrinkles artificially modify the extracted value of the fracture toughness, an effect not accounted for in our uncertainty analysis. Although the transfer was performed in such a way as to avoid producing wrinkles in graphene, the removal of the underneath layer can also introduce wrinkles in graphene.^{Erreur ! Source du renvoi introuvable.} A wrinkled freestanding graphene tends to self-fold under deformation. This is indeed what has been observed for the widest specimens, as a way to release the in-plane strain energy. In this work, the presence of wrinkles near the crack tip is unlikely, especially in all the 80 measured specimens (see Supplementary Material VI for more details). For asymmetric design, creases have been sometimes observed and can be partially responsible for the (artificial) fracture toughness variation '

For TOC results,

‘The discrepancy in the TOC results can be related to the presence of wrinkles. As mentioned earlier, the quantification of the impact of wrinkles on the extracted Young’s modulus is a challenging problem that has been studied numerically. Shen *et al.*^{Erreur ! Source du renvoi introuvable.} showed a negligible impact of wrinkles on the stiffness of the monolayer graphene layer. In the worst-case scenario, high wrinkle density with high wavelength involves an 11% reduction of Young’s modulus along the armchair direction. About 4% reduction along a zigzag direction is in the range of the measurement error on the Young’s modulus determined in this study. This value could be even lower since the wrinkle density in the present work is low. On the other hand, Qin *et al.*^{Erreur ! Source du renvoi introuvable.} reveal higher strength in the case of wrinkled graphene specimens compared with flat ones, which was used as an argument to strengthen some metal matrix composites.^{Erreur ! Source du renvoi introuvable.} Consequently, the extracted Young’s modulus will not change drastically in the presence of a small wrinkle’s amplitude. Another point worth mentioning is that the results obtained here are all consistent with one another and when a test is repeated, even though corrugations are not uniformly distributed. This consolidates the trust in the validity of the results (more details are provided in Supplementary Material VI). Note finally that the main interest of the TOC structures was not to look at the elastic behaviour but mainly to determine the fracture strain and corresponding fracture stress on a large set of specimens.’

Also, in order to more extensively cover the possible limitations and artefacts in our tests, new supplementary material has been added to the revised version, basically summarizing the responses we have given here above, as well as the other sources of imperfections (wrinkles, creases, PMMA residues) addressed in further comments. This supplementary section is named “On the sources of imperfection/pollution in the graphene on-chip tests”

Comment 2. How does the fail-safe fracture toughness reported compare to other relevant engineering materials?

The short answer is: “very poorly” if one compares it with tough steels, nickel alloys, or high entropy alloys, which show K_{Ic} sometimes above $250\text{MPa}\sqrt{\text{m}}$. However, graphene performs extremely well when compared with hard thin film systems in which K_{Ic} rarely exceeds $2\text{MPa}\sqrt{\text{m}}$. So it is either “very good” or “very bad”.

As a matter of fact, this a complicated subject that relates also very much to the thickness dependence of fracture toughness. As the thickness decreases, G_{Ic} generally tends first to increase when entering the so-called plane stress regime and then decreases, ultimately down to values that should at some point reach twice the surface energy when dissipation mechanisms become rare or difficult to operate. This is true for metallic materials (and this is, incidentally, a core research topic of one of the co-authors, e.g. Pardoen *et al.*, J Mech. Phys Solids 2004) but also for more brittle materials.

In practice, the graphene fracture toughness outperforms any hard thin layer of any oxide, carbide, boride, or nitride film in the sub-100 nm range. Only DLC layers show similar levels of K_{Ic} (but generally for thicker films in the 1-micrometer range, e.g. Jonnalagadda, JMPS 2008) but it is not clear how the K_{Ic} will eventually decrease when decreasing the DLC thickness.

We have added one sentence in the conclusion regarding this complex point.

This fracture toughness value is higher than any hard thin layer in the sub-100 nm range although still very low when compared with bulk tough steels or high entropy alloys with values above $250 \text{ MPa}\sqrt{m}$ because energy dissipation at the crack tip is limited by the thickness.

Comment 3. Does any residue remain on the graphene after transfer? How is it ensured that this is removed? Does any residue influence the mechanical measurements?

The PMMA residues can be present in some parts of graphene but only in very small areas. A SEM investigation is carried out after each graphene transfer to make sure that no residues remain in the area of interest at least in terms of detectable size residues. For very small residues, we cannot provide firm arguments to exclude them. The only thing we can say is that the combination of annealing along the use of hot acetone helps in getting rid of PMMA residues (e.g. Hwangbo *et al.*, Scientific Reports 2014). It has been shown in the literature that PMMA-based graphene transfer produces a continuous film of residues with a thickness of 1 to 2 nm or some sort of PMMA islands. Now, in this work, we believe that we do not have a continuous film of PMMA but more some islands of residues with low density thanks to using hot acetone. Hwangbo *et al.* (Sci. Reports 2014) demonstrate that the bonding strength between graphene and the PMMA residues is very weak. Therefore, synergetic toughness enhancement of the graphene by residues even in the case of a uniform layer of PMMA is unlikely. The fracture toughness will not be impacted by the presence of some islands of residues especially since the fracture toughness of PMMA is low around 1 to 2 nm. However, the cracking rate and path can be influenced in the case of thick residues that can behave as crack arrestors.

Now, if these minor residues would still be attached to some specimens, one can reasonably assume that they will not have a major impact on the mechanical behavior of graphene. This is also the interest of performing a large number of tests to minimize the impact of some imperfect specimens.

As a final comment, it is worth mentioning, that the step that introduces more PMMA residues is not the graphene transfer but the lithography step. This step used PMMA underneath the photoresist since residues of PMMA are easier to remove than photoresist. The removal of PMMA in this step is based on hot acetone rinses leaving a layer of resist residues. Moreover, characterizing graphene with PMMA residues is still interesting since the fabrication of graphene-based transistors and logic circuits strongly relies on PMMA as well.

This aspect is discussed in the new supplementary materials added to the revised version and called from the paper when listing the sources of uncertainty/imperfections.

‘Another possible artifact could be due to PMMA residues on graphene. Now, the bonding strength between PMMA residues and graphene is weak besides the fact that the K_{Ic} of PMMA residue is low $\sim 1 \text{ MPa}\sqrt{m}$.^{Erreur ! Source du renvoi introuvable.} Hence, PMMA residues islands can potentially affect the cracking rate and path but not the fracture toughness. Here, the cracking path was in most cases similar from one specimen to another excluding thus any significant impact of residues on K_{Ic} .’

Referee 2

General comment. In this manuscript, the authors report a very comprehensive experimental effort to measure fracture strength and toughness of graphene, through a microfabricated platform that allows uniaxial and notched-specimen testing of many samples. The authors report 80 results for the toughness of graphene, and about ~50 (this reviewer's estimation) data for strength. As such, this paper represents an unprecedented effort to characterize these quantities with statistical significance. The experiments themselves are a praiseworthy achievement. The toughness reported is generally in line with previous reports, although the amount of data gives greater confidence to this result. The maximum strength reported also falls in line with expectations, but the variation with size is nicely shown experimentally.

I recommend publication of this manuscript after the following major revisions are addressed.

Thank you very much for your positive opinion of our study. Actually, we recorded 222 TOC successful specimens not failing under loading. If we count the broken ones, the number is much higher.

Comment 1. The authors should explain better two aspects of the notched test methodology: i) Although many details of the fabrication are given, I could not see explicitly stated how the initial notch is fabricated. Is it lithographically patterned? I understand a sharp crack emanates from this notch, which later arrests itself at a given length. ii) Please add a succinct explanation of the K_{Ic} calculation in the main text. It is explained in detail in the supplementary material, but a short explanation with the main gist should be given in the main.

Due mainly to the restrictions on the number of words, we did not include many details regarding different key elements in the original version of the paper, involving K_{Ic} extraction and other important experimental issues. But we agree that this is necessary for carrying a more convincing message and we have provided more information now in the core of the text.

The initial notch is indeed fabricated by a photolithography technique similar to the rest of the test structure and test frame as mentioned in the Methods section.

More details of both on-chip techniques were added in the revised version such as:

“In an effort to overcome the shortcomings encountered in the available approaches, several challenges have been addressed in this study: (i) the specimens were tested on-chip to circumvent the gripping, clamping, and transfer problems; (ii) the specimen shape was accurately controlled through lithography methods and the geometrical dimensions can be measured with precision; (iii) an on-chip loading relying on a residual stress actuation principle was adapted to avoid the use of any external macroscopic or microscopic device as well as the associated alignment issues; (iv) many samples were produced and tested simultaneously for statistical analysis; (v) different sample sizes and shapes were processed to verify if the size dependence of the fracture resistance related to the defects population can be rationalized; (vi) the crack arrest principle adopted to circumvent the artifact of a blunted starter notch instead of a true pre-crack in the fracture mechanics sense required an adapted design.”

“In the case of the TOC configuration, the imposed displacement is generally applied to a dogbone specimen while in the case of the COC configuration, the displacement is applied to a notched specimen to generate a crack from the tip of the notch. The working principle of the

combination of the COC and TOC configurations into the new TOCOC test chain to determine the representative flaw size responsible for failure is illustrated in **Erreur ! Source du renvoi introuvable.**

The specimen is deformed owing to the contraction of the attached actuator beam reaching a stable position corresponding to force equilibrium. The displacement is measured by SEM between movable and fixed cursors as shown in **Erreur ! Source du renvoi introuvable.**

Each deformed specimen represents a single point on the stress-strain diagram. Thus, several TOC structures must be fabricated with different specimen and actuator beam lengths in order to vary the applied stress level covering the range from small elastic deformations up to fracture.

More information on the extraction of K_{Ic} is incorporated in the main text:

The magnitude of K_I can be roughly estimated in most cases by the following expression (see Supplementary III for more details):

$$K_{I_{DCB}} = (1 - \nu_a) \sigma_a^{\text{int}} \sqrt{L_a} \frac{4 \sqrt{\frac{6L_a}{\alpha_2 L_s}}}{32 \frac{E_a}{E} \frac{a^2}{L_s^2} + \frac{L_s}{a} \frac{L_a}{W_a} \frac{t}{t_a}}, \quad (1)$$

where ν_a is the Poisson ratio of the actuator, ν is the Poisson ratio of the test specimen, here graphene, $\alpha_2 = 1 - \nu^2$ in plane strain and $\alpha_2 = 1$ in plane stress, σ_a^{int} is the residual stress in the actuator prior release, L_a is the actuator length, L_s is the specimen length, E_a is Young's modulus of the actuator, E is Young's modulus of the specimen, t_a is the actuator thickness, t is the specimen thickness, and W_a is the actuator width. The fracture toughness K_{Ic} can be estimated from equation (1) by replacing the crack length a with the final crack arrest length a_{arrest} . But, once again, FE simulations have been systematically performed to extract more accurate values for the stress intensity factor.

From equation (1), one can conclude that the stress intensity factor K_I is proportional to the residual stress in the actuator and to its length, with a dependence on the crack length and on several other geometrical quantities. The parameters required to perform the FE simulations were determined experimentally as given in Table S1 with related uncertainties”

We have added some extra information regarding the notch:

“The fracture toughness is extracted from the final crack arrest length a_{arrest} , solving the problem of producing extremely sharp pre-cracks and associated artifacts, such as the notch blunting effect at crack initiation. Benefiting from the lithography process to induce the pre-crack avoids the damage produced while using a focused ion beam as in the few prior studies on the subject.”

More details regarding the TOCOC method are also added in the revised version:

“In the TOCOC method, the strength of graphene σ_c obtained from TOC structures is linked to the fracture toughness K_{Ic} determined by COC structures based on $a_c = \frac{1}{\pi} \frac{K_{Ic}^2}{\sigma_c^2}$ which provides the critical defect size a_c responsible for triggering the failure of graphene-based devices as schematically explained in **Erreur ! Source du renvoi introuvable.**”

Comment 2. Please explain the influence of creasing in the tensile testing results. A thin film stretched as in the dogbone or rectangular samples will unavoidably develop longitudinal wrinkles or creases. I expected the finite-element simulations or theoretical model to address this, but it is not included. It is tangentially mentioned in the manuscript as "twisting". The authors should show this effect does not significantly influence the results. Creasing can lead to observed stiffening, which is a structural effect, and not an intrinsic property of the material.

The question of creasing is a difficult one, similar to the question about the wrinkles already addressed when answering Comment 1 of Referee 1. We invite Referee 2 to read this response, which has some close connection with the present point. But, we understand that the Referee points more here towards the creases that are generated when deforming extremely thin specimens due to combinations of imperfections and misalignments in the geometry, and the Poisson effect impeded at the clamping area (as a structural effect). We take it as a different instability phenomenon and differentiate thus the terminology “crease” and “wrinkle”, although, maybe, some could argue that a crease is a class of wrinkles (Two of the co-authors, Pugno and Pardoen, have been working – independently - on wrinkling but we have limited experience on creases).

The key point is that we have seen creases only in the asymmetric COC specimens (cracked specimen), as observed in Figure 2. However, the symmetric specimens, owing to more constraint put on the test structure, as well as the TOC (uniaxial specimens) do not show significant creasing upon deformation – probably the creases are favored by the presence of the crack. We have been running 3D FE simulations with Abaqus with imperfections, but not with the goal of generating creases and studying them, but to estimate the mixed mode effects on cracking. The analysis was not pushed further because the effects were found weak. These simulations were further pursued by a team from the University of Twente who delivered convincing data. The fact is that these 3D simulations never lead to creasing. This is often the case with such types of instabilities that one needs to seed the right imperfection to generate it. This appears to be a research project on its own that goes beyond what can be achieved in the context of this study.

We have added the following comment:

“For asymmetric design, creases have been sometimes observed and can be partially responsible for the (artificial) fracture toughness variation.”

Comment 3. The grain boundary size is presented somewhat ambiguously. The authors seem to suggest they test mono-crystal graphene in most cases because the grain size is ~3 cm, supported by Figure S3. This figure is a photograph of graphene on copper, so I am puzzled as to why this picture demonstrates large grains. Furthermore, Figure S2 shows a grain size of ~200 μm . The toughness results probably are not affected by this (sample size ~10 μm) but the large tensile samples may have some boundaries. This is not necessarily a problem. These

experiments are difficult. But if boundaries are potentially present, this should be acknowledged.

This is indeed an important comment. The micrograph on top of Cu shows a grain domain of 3 cm. This Cu/graphene group undergoes oxidation in order to highlight the grain boundaries, the Cu part that is covered by graphene will not oxidize and its color will not become reddish since it is protected by graphene while if there is any grain boundary in the graphene part of Cu will not be covered hence will oxidize. This sample is a witness sample since the used sample cannot undergo this oxidation step to not degrade the graphene properties. The same recipe is used assuming it will lead to the same grain size, also SEM verification is made to check if the grown graphene is continuous or not. Fig. S5 shows separated domains of around 200 μm . This was presented only to show that depending on the recipe and precisely the growth time, we can go from non-continuous graphene to continuous, large-domain graphene. However, we confirm that the graphene specimens tested in this work are continuous, coming from cm-sized domains. Therefore, the probability of having grain boundaries is extremely low and this is why we disregarded this aspect. Moreover, sample sizes are small and TOC samples are smaller than COC. Note finally that the largest TOC sample is 40 μm -long and 8 μm -wide while the largest COC sample is 10 μm -long and 90 μm -wide. Another element to mention, the size effect on fracture toughness was shown for very small grain sizes around 250 \AA – thus, even if by bad luck one would have one GB entering one of our COC specimens, the impact would remain small.

Comment 4. Optional. The Weibull analysis could be refined. The authors are reporting different m exponents for different sizes. Bazant and others have shown that a given material has the same m , and the size is what causes the variation in the distributions. In other words, strictly speaking, using different m 's for the same material is incorrect, although it can give a workable description of the results. The authors could consider using the methods from Mechanics of Materials 162, 104057, 2021, which can allow the calculation of parameters from the strength data that could be compared with defect size and theoretical strength, to extract further insights from their data.

We have slightly modified the writing to clarify the message. The fact that “ m ” is changing is due to an additional class of defects that appear when changing the specimen size or morphology (such as tubing or warping effects). Only the smallest specimens include only one class of defects that we claim to be 5/7 pairs. Thank you very much for the very interesting reference that is cited in the revised version. One important aspect concluded from the paper is to not group samples with different dimensions in the Weibull analysis that we applied in the revised version resulting in m similar to what was reported in the literature.

‘The rectangular specimens are characterized by a high value of Weibull modulus $m = 27$ while the dogbone specimens exhibit a lower value of m as detailed in **Erreur ! Source du renvoi introuvable.** Lower m are obtained for larger specimens indicating that another population of defects/imperfections is playing a role typically due to the twisting of the specimens. Hence, one can hardly rationalize these data into one single master plot. **Erreur ! Source du renvoi introuvable.**’

‘The variation of m particularly between the dogbone and rectangular graphene samples is mainly attributed to the size and geometry differences and not to the material property since m is the same for a given material. For instance, wider samples tend to remain more in-plane compared with the longer and narrower samples that likely folded acting like nanotubes more than a flat sheet. In another vein, the variation of Weibull moduli in this work reflects the sensitivity of CVD-monolayer graphene to the

presence of defects such as creases that can deteriorate under cyclic loading as confirmed by Cui et al.¹ Erreur ! Source du renvoi introuvable. One more note, the high value of m indicates a high quality of graphene especially for rectangular samples. ’

Referee 3

General comment. This paper reports an on-chip method used to measure fracture toughness, stress, and strain of monolayer graphene. Residual stresses in deposited films are used to mechanically load graphene, and the final strain is measured by SEM. Many samples are measured to provide high statistical confidence in the fracture properties. Analysis of the results led to the insightful conclusion that 5-7 defect structures, having 1.4 nm length and prevalent in even high-quality graphene samples, are the limiting factor in graphene's fracture strength.

The experiments reported are clever, challenging, and provide greatly improved bounds on fracture properties of graphene. The statistical analysis is well explained and the discussion provides thoughtful insights. I provide several minor comments below, but overall find the work to be of high quality and interest. I recommend it for publication after minor revision.

This is a very motivating comment – thank you.

Comment 1. Work on suspended graphene drums has reported pre-strain of the graphene when transferred/pinned to a solid wafer. Can you comment on whether pre-strain may be present in your graphene films and what effect this would have on the measured fracture strain, etc.?

In this first study, we indeed neglected the residual stresses due to the lack of a study that adequately quantifies the intrinsic residual stress in graphene. One justification is that we assumed the impact was negligible with respect to the extremely high-stress levels that are attained in the test. Motivated by the comment, we have challenged this tacit assumption.

López-Polín *et al.* (Carbon, 2017) found a pre-tensile stress value for monolayer exfoliated suspended graphene drums, with diameters ranging from 0.5 to 3 μm , in the range of 0.05 to 0.6 N/m. This value is close to the values given by AFM indentation performed by Lee *et al.* and earlier reported by López-Polín *et al.* (Nat. Phys. 2015). Based on these residual stress levels, we re-analyzed the data using the scheme established for the on-chip tests which integrate the presence of residual stress in the test specimens. This leads to a change in the estimated fracture strain and fracture stress not exceeding 1%, justifying the assumption.

We have added a comment in the revised version explaining the absence of a significant impact of residual stress.

“The results shown in Fig. 5 were obtained without considering any pre-tensile stress in the graphene. By considering pre-tensile stress levels of 0.05 to 0.6 N/m, the fracture strain and stress could change by about 1% justifying the assumption of neglecting the graphene's pre-strain.”

Comment 2. The release/deformation process is said to occur fast (L298). Is it possible that there will be overshoot in this process, providing higher peak stress and strain than recorded in the SEM in the final settled state?

The actuator beams are designed to be wider than the test specimens to ensure specimens are released first. The geometry of the actuators is also tapered so that loading starts when the part next to the overlap with the specimen is released first and ends when the wider part of the actuator beam (at the anchor) is released. The strain rate during the loading of the specimen is controlled by the etch rate of the underlying silicon and the angle of the tapering of the actuator.

The etching time of silicon in XeF₂ being short and the tapering angle being small the release process is indeed considered fast but still occurs progressively.

The sentence mentioning this release process was edited to clearly explain the abovementioned point.

“(which occurs progressively thanks to the tapering of the actuator beams until the release of the last attached point, which can be considered as a fast release).”

Comment 3. While most PMMA is removed from graphene by acetone, residue usually remains. Will this have any effect on your measurements?

The PMMA residues can be present in some parts of graphene but only in very small areas, a SEM investigation is carried out after each graphene transfer to make sure that no residues remain in the area of interest at least in terms of detectable size residues. For very small residues, we cannot provide firm arguments to exclude them. The only thing we can say is that the combination of annealing along the use of hot Acetone helps in getting rid of PMMA residues (e.g. Hwangbo *et al.*, Scientific Reports 2014). It has been known in the literature that PMMA-based graphene transfer produces a continuous film of residues with a thickness of 1 to 2 nm or some sort of PMMA islands. Now, in this work, we believe that we do not have a continuous film of PMMA but more some islands of residues with not high density thanks to using hot acetone. Hwangbo *et al.* (Sci. Reports 2014) demonstrate that the bonding strength between graphene and the PMMA residues is very weak. Therefore, synergetic toughness enhancement of the graphene and residues even in the case of a uniform layer of PMMA is unlikely. The fracture toughness will not be impacted by the presence of some islands of residues especially since the fracture toughness of PMMA is low around 1 to 2 nm. However, the cracking rate and path can be influenced in the case of thick residues that can behave as crack arrestors.

Now, if these minor residues would still be attached to some specimens, one can reasonably assume that they will not have a major impact on the mechanical behavior of graphene. This is also the interest of performing a large number of tests to minimize the impact of some imperfect specimens.

As a final comment, it is worth mentioning that the step that introduces more PMMA residues is not the graphene transfer but the lithography step. This step used PMMA underneath the photoresist since residues of PMMA are easier to remove than photoresist. The removal of PMMA in this step is based on hot acetone rinses leaving a layer of resist residues. Moreover, characterizing graphene with PMMA residues is still interesting since the fabrication of graphene-based transistors and logic circuits strongly relies on PMMA as well.

Comment 4. The finite element analysis and the stress intensity equations appear to rely on the assumption of an isotropic continuum material. However, the measurements are of monocrystal, monolayer graphene. Can you please comment on the validity, or any error introduced in these calculations as a result?

Based on the elastic constants characterizing the in-plane linear elastic response of graphene C_{11} (358 N/m) and C_{12} (60 N/m) from Wei *et al.* in Physical Review 2009 (thus neglecting any higher-order elasticity effects), one can calculate the elastic modulus as a function of the direction of loading using basic elasticity relationships. The modulus varies between 341 and 358 N/m which means less than 5% between the maximum and minimum stiffness. Graphene

is weakly elastically anisotropic when considering in-plane loadings. Such effects are within the experimental uncertainty.

We have added a comment in the revised version:

“The FE simulations and the uncertainty study did not account for graphene’s anisotropic elastic behavior. As a matter of fact, the elastic modulus when determined for different in-plane loading directions using the elastic constants $C_{11} = 358$ N/m and $C_{12} = 60$ N/m (from reference^{Erreur ! Source du renvoi introuvable.}) involves less than 5% variation between maximum and minimum stiffness which is within experimental uncertainty.”

Comment 5. I suggest providing more of an explanation of the chip structure and how the chip test works in the main paper. Although the methods and supplementary provide more detail, it would be easier to follow with a little more explanation up front. Lines 81-95 reference Fig. 1 a few times for details, including the cursors and how a crack propagates, that are not clearly labelled. Fig. S1 provides more detail, but both figs would benefit from more words in the caption, labels in the panels, and text describing the fig/procedure.

Thanks for pointing it out, the revised version was changed to include more details regarding the working principle of on-chip. Please check the answer given to Referee 2 for comment 1.

Fig. 1 and Fig. S1 are modified and now show the location of both cursors; more details were added to the captions as well.

Comment 6. I find the conclusion that 5-7 defects limit the fracture strength of graphene to be quite insightful and supported by the analysis. The point made that small enough areas of graphene could possibly be free from such defects and survive larger strains is also interesting. However, I find that the way this is referenced in the abstract is unclear before reading the full paper (“Except for micron-sized membranes, which could be designed based on 110 GPa strength, fail-safe design should be based on 57 GPa strength”). I suggest revising this sentence of the abstract to make the point clearer to anyone who only reads that far.

Thanks for the positive feedback. The sentence has been revised:

“Micron-sized graphene membranes can be produced defect-free and the design rule can be based 110 GPa strength. For larger areas, a fail-safe design should be based on 57 GPa strength level. “

Comment 7. Some variables in Fig. 1 / Table1 seem to not be defined until later in the paper. Would be good to define in text or captions.

Good idea, we added the definition of each parameter in Fig. 1 / Table 1.

Comment 8. The reference numbers in Fig. 2b seem to be wrong.

You are right. Thanks for telling us, this has been corrected in the revised version.

Comment 9. Please state in the caption what the error bars in Fig. 2b show. It appears to be standard deviation or something similar because the bar with only one sample has no error

bars. I suggest also noting that the error bars for this one are not shown/calculated because there was one sample, or otherwise estimating the measurement uncertainty.

We would to thank you for this remark. The error bars in Fig. 2b refer to standard deviation. We added an explanation regarding this point in the caption and we removed the error bar on one of the experiment including only one specimen.

Comment 10. Fig. 5a&b the numbers in the legend are not explained until the methods section. I suggest explaining them in the main text or caption.

Thank you for this comment, more details are now given in the legend of Fig. 5a&b.

REVIEWERS' COMMENTS

Reviewer #1 (Remarks to the Author):

The authors have adequately addresses the concerns raised. The manuscript is now suitable for publication.

Reviewer #2 (Remarks to the Author):

This manuscript is the second round of review. At this point, I believe the authors satisfactorily addressed my comments, and have explained in the text several experimental challenges, which allows the reader to gain more context about the results. I recommend publication at this point.

Reviewer #3 (Remarks to the Author):

The authors have addressed all my comments. I recommend this paper for publication.